# CLASP: An online learning algorithm for Convex Losses And Squared Penalties

## Abstract

We study Constrained Online Convex Optimization (COCO), where a learner chooses actions iteratively, observes both unanticipated convex loss and convex constraint, and accumulates loss while incurring penalties for constraint violations. We introduce CLASP (Convex Losses And Squared Penalties), an algorithm that minimizes cumulative loss together with squared constraint violations. Our analysis departs from prior work by fully leveraging the firm non-expansiveness of convex projectors, a proof strategy not previously applied in this setting. For convex losses, CLASP achieves regret $O\left(T^{\max\{\beta,1-\beta\}}\right)$ and cumulative squared penalty $O\left(T^{1-\beta}\right)$ for any $\beta \in (0,1)$. Most importantly, for strongly convex problems, CLASP provides the first logarithmic guarantees on both regret and cumulative squared penalty. In the strongly convex case, the regret is upper bounded by $O(\log T)$ and the cumulative squared penalty is also upper bounded by $O(\log T)$.

## 1 Introduction

We consider a setting where at each iteration $t \in \{1, 2, 3, \dots\}$, a learner selects an action $x_t$ from a convex set $\mathcal{K} \subset \mathbb{R}^n$, then a loss function $f_t$ is revealed, and the learner incurs loss $f_t(x_t)$. The learner's goal is to perform nearly as well dynamically as the best fixed action in hindsight. i.e, keep the regret

$$\text{Regret}_T = \sum_{t=1}^{T} f_t(x_t) - \sum_{t=1}^{T} f_t(x_T^\star). \tag{1}$$

growing sublinearly in $T$, so that, asymptotically, the average loss of the learner is no worse than the optimal fixed action, $x_T^\star = \arg\min_{x \in \mathcal{K}} \sum_{t=1}^{T} f_t(x)$.

Online Convex Optimization (OCO) is the special case of Online Learning in which the action set $\mathcal{K}$ is convex, and the loss functions $f_t$ are convex. The OCO setup has been extensively studied over the years (Shalev-Shwartz et al., 2012; Hazan et al., 2016; Zinkevich, 2003; Duchi et al., 2011; Bubeck et al., 2012; Hazan & Kale, 2012; Hazan & Minasyan, 2020; Hazan & Singh, 2021). OCO assumes that operational constraints on the possible actions are static in time, being fully captured by the fixed set $\mathcal{K}$.

In many applications, however, operational constraints do change at each iteration: at time $t$, the action $x_t$ must satisfy not only $x_t \in \mathcal{K}$ but also an extra constraint $g_t(x_t) \leq 0$, with $g_t$ a convex function. The challenge here is that the learner must choose an action $x_t$ also *before* knowing the constraint function $g_t$. This more difficult setup generalizes OCO and is known as Constrained Online Convex Optimization (COCO).

To conclude, the COCO learner wishes not only to bound the regret (1), now with

$$x_T^\star = \arg\min_{x \in \mathcal{K} \cap \mathcal{C}_T} \sum_{t=1}^{T} f_t(x), \quad \mathcal{C}_T = \bigcap_{t=1}^{T} C_t, \quad C_t = \{x : g_t(x) \leq 0\}, \tag{2}$$

but also wishes to bound some notion of cumulative constraint violation (CCV), as detailed in the next section.

## 1.1 RELATED WORK

Early work on COCO (Mahdavi et al., 2012; Jenatton et al., 2016; Yuan & Lamperski, 2018; Yu & Neely, 2020) focused on soft constraint violations, allowing the learner to compensate over time by balancing positive and negative violations. In those settings the performance is measured by the cumulative constraint violation $\text{CCV}_T = \sum_{t=1}^{T} g_t(x_t)$. However, this metric has a critical weakness: The sum may become negative even if many rounds incur positive violations, thereby masking the severity or frequency of actual constraint breaches.

Such arithmetic compensation is often unsuitable in practice, as many applications demand a more direct and monotonic measure of constraint violation. We thus focus on the hard metrics

$$\text{CCV}_{T,1} = \sum_{t=1}^{T} g_t^+(x_t), \quad \text{or} \quad \text{CCV}_{T,2} = \sum_{t=1}^{T} \left( g_t^+(x_t) \right)^2, \tag{3}$$

where $g_t^+(x) = \max\{0, g_t(x)\}$. Under these definitions, violations cannot be offset over time: these cumulative sums are nondecreasing in $T$. Both metrics are of practical relevance, and authors generally commit to one as their principal violation measure. A natural question is whether a bound on $\text{CCV}_{T,1}$ can be turned into one on $\text{CCV}_{T,2}$ by redefining $g_t$ as $(g_t^+)^2$. This reduction, however, is generally invalid: key assumptions often required for $\text{CCV}_{T,1}$ bounds—such as the existence of a Slater point $\tilde{x}$ with $g_t(\tilde{x}) < 0$ for all $t$—cannot hold for $(g_t^+)^2$, which is always nonnegative. Other structural properties (e.g., convexity or Lipschitz constants) may also be lost under squaring. For these reasons, we directly analyze $\text{CCV}_{T,2}$ rather than rely on such a transformation. Moreover, even if the reduction were possible, it would not yield our main result: to the best of our knowledge, no $O(\log T)$ bound exists for $\text{CCV}_{T,1}$ in the strongly convex setting, so our logarithmic $\text{CCV}_{T,2}$ guarantee cannot be inferred from prior $\text{CCV}_{T,1}$ results.

We now review the closest work in COCO, organized by whether the constraints are **static** ($g_t = g$ for all $t$) or **dynamic** ($g_t$ may vary adversarially). Our focus is on the dynamic case, but it is useful to contrast with the static regime first.

**Static constraints.** As the focus of this paper is on dynamic constraints, we restrict ourselves here for brevity to convex loss functions, omitting the results for strongly convex ones. The early work of (Mahdavi et al., 2012) uses a regularized Lagrangian update and achieves $\text{Regret}_T \leq O(\sqrt{T})$ and soft violation $\text{CCV}_T \leq O(T^{3/4})$. (Jenatton et al., 2016) refine this approach via adaptive weightings of primal and dual step sizes, obtaining $\text{Regret}_T \leq O(T^{\max\{\beta, 1-\beta\}})$ and $\text{CCV}_T \leq O(T^{1-\beta/2})$ for any $\beta \in (0, 1)$. (Yu & Neely, 2020) push further by bounding $\text{CCV}_T \leq O(1)$ (constant violation) while preserving $O(\sqrt{T})$ regret. Yi et al. (Yi et al., 2021), in turn, address *hard* violations by showing $\text{Regret}_T \leq O(T^{\max\{\beta, 1-\beta\}})$ and $\text{CCV}_{T,1} \leq O(T^{(1-\beta)/2})$. Finally, (Yuan & Lamperski, 2018) focus directly on the squared-penalty metric $\text{CCV}_{T,2}$, proving $\text{Regret}_T \leq O(T^{\max\{\beta, 1-\beta\}})$ and $\text{CCV}_{T,2} \leq O(T^{1-\beta})$. Their work penalizes large violations more heavily, preventing cancellation that occurs under soft violation metrics.

**Dynamic constraints.** When the constraints $g_t$ may vary arbitrarily with $t$, the problem becomes significantly more challenging.

(Guo et al., 2022) introduce the Rectified Online Optimization (RECOO) algorithm, which exploits a first-order approximation of the regularized Lagrangian at each round. For convex losses, RECOO achieves $\text{Regret}_T \leq O(\sqrt{T})$ and $\text{CCV}_{T,1} \leq O\left(T^{\frac{3}{4}}\right)$. In the strongly convex case, it improves to $\text{Regret}_T \leq O(\log T)$ and $\text{CCV}_{T,1} \leq O(\sqrt{T \log T})$.

(Yi et al., 2023) extend these ideas in a distributed setting, with results that can be transposed to the centralized regime. For convex losses, they establish $\text{Regret}_T \leq O\left(T^{\max\{\beta, 1-\beta\}}\right)$ and $\text{CCV}_{T,1} \leq \left(T^{1-\beta/2}\right)$ for any $\beta \in (0, 1)$. For strongly convex losses, their guarantees become $\text{Regret}_T \leq O\left(T^{\beta}\right)$ and $\text{CCV}_{T,1} \leq \left(T^{1-\beta/2}\right)$.

A major step forward is due to Sinha & Vaze (2024), who introduced the Regret Decomposition Inequality, an elegant analytical tool that extends the drift-plus-penalty framework (Neely, 2010).

They show that suitably modified AdaGrad variants can achieve $\text{Regret}_T \leq O\left(\sqrt{T}\right)$ and, for the first time, $\text{CCV}_{T,1} \leq O\left(\sqrt{T}\log T\right)$ for convex loss functions. For strongly convex functions, the results are $\text{Regret}_T \leq O\left(\log T\right)$ and $\text{CCV}_{T,1} \leq O\left(\sqrt{T\log T}\right)$, with the CCV bound improved to $\text{CCV}_{T,1} \leq O(\log T)$ for the specialized setting of non-negative regrets.

Building on this line Vaze & Sinha (2025) propose the *Switch* algorithm, which explicitly leverages the geometry of the constraint sets $C_t$ (2). Although the strongly convex case is not explicitly addressed, for convex loss functions Switch guarantees $\text{Regret}_T \leq O(\sqrt{T})$ and $\text{CCV}_{T,1} \leq O(\sqrt{T}\log T)$ across all instances, the novelty being that the bound on CCV can drop to $O(1)$ for special geometrical instances of $C_t$. Our proposed algorithm, CLASP, bears certain similarities with Algorithm 2 in (Vaze & Sinha, 2025), as we elaborate in the next section.

To conclude, we highlight an interesting line of COCO research on low-complexity, projection-free algorithms, which avoid projection onto $\mathcal{K}$. Representative work is (Garber & Kretzu, 2024; Sarkar et al., 2025; Wang et al., 2025; Lu et al., 2025). For example, the recent work (Lu et al., 2025) exploits a separation oracle to achieve $\text{Regret}_T \leq O\left(\sqrt{T}\right)$ and $\text{CCV}_{T,1} \leq O\left(\sqrt{T}\log T\right)$ for convex loss functions; and $\text{Regret}_T \leq O\left(\log T\right)$ and $\text{CCV}_{T,1} \leq O\left(\sqrt{T\log T}\right)$ for strongly convex ones.

## 1.2 CONTRIBUTIONS

We propose **CLASP**, an online COCO algorithm for dynamic constraints with Convex Losses And Squared Penalties, where the violation metric of interest is $\text{CCV}_{T,2}$ (3). Our contributions can be summarized as follows:

- **Strongly convex losses.** CLASP achieves $\text{Regret}_T \leq O(\log T)$ and $\text{CCV}_{T,2} \leq O(\log T)$. To the best of our knowledge, this is the first result to guarantee logarithmic bounds on *both* regret and squared violations. The closest prior work, (Sinha & Vaze, 2024), establishes $\text{Regret}_T \leq O(\log T)$ and $\text{CCV}_{T,1} \leq O(\log T)$, but only in the restricted setting of non-negative regrets. CLASP removes this limitation by providing guarantees without assuming the sign of the regret. Furthermore, by the Cauchy-Schwarz inequality, we have

$$CCV_{T,1} = \sum_{t=1}^{T} (g_t)^+(x_t) \leq \sqrt{T}\sqrt{CCV_{T,2}} \leq O(\sqrt{T\log T}),$$

  thus CLASP also attains the best-known bound for $CCV_{T,1}$ in the strongly convex setting.

- **Convex losses.** For general convex losses, CLASP guarantees

$$\text{Regret}_T \leq O(T^{\max\{\beta, 1-\beta\}}), \quad \text{CCV}_{T,2} \leq O(T^{1-\beta}), \qquad \forall \beta \in (0, 1).$$

  This matches the rates reported by Yuan & Lamperski (2018), whose results were derived for static constraints; CLASP extends them to the dynamic regime and within the same analysis framework, unifying the results under a single line of reasoning.

- **Algorithmic simplicity.** Each CLASP iteration consists of a gradient step with respect to the latest loss, followed by a single projection onto the current feasible set $\mathcal{K} \cap C_t$. In contrast, Algorithm 2 of Vaze & Sinha (2025) requires two projections per iteration, one onto the full historical intersection $\mathcal{K} \cap \bigcap_{\tau=1}^{t} C_\tau$. Thus CLASP is more memory-efficient, avoiding the need to retain past constraints. Moreover, the analyses of the two methods differ fundamentally, as discussed in Sections 3.2 and 4.

- **Analytical novelty.** Our analysis relies on the *firm non-expansiveness* (FNE) of projections onto closed convex sets—a stronger property than the standard non-expansiveness typically used. Exploiting FNE allows a clean modular proof structure, where regret and constraint violation are analyzed separately. This modularity also makes the analysis more extensible, e.g., to multiple dynamic constraints or persistent constraints (Section 4).

**Relevance to Machine Learning Practice.** Recent empirical and theoretical work suggests that strong convexity and sharper penalties for violations are practically meaningful in constrained ML settings. For example, Wang et al. (2025) shows that under strong convexity of regret, both regret and constraint violations drop substantially, and experiments in that paper confirm that algorithms

exploiting strong convexity perform better on real datasets. The authors of Ma et al. (2025) empirically validate that in resource allocation problems with hard constraints plus regularization, faster rates (logarithmic regret) are achievable. The work Banerjee et al. (2023) finds that many practical deep models satisfy restricted strong convexity, and fast (geometric) convergence is observed in training. These works show that assumptions like strong convexity are not merely theoretical and that penalties, and constraints (Ramirez et al. (2025)), matter in real ML systems. However, prior work usually measures constraint violation linearly (or counts violations) rather than providing log-rate guarantees on both regret and the *severity* of violations (e.g., squared penalties). The squared penalty can be useful in applications where large violations of the constraints are a reason for concern. For example, in model predictive control, the inequality $g(x) \leq 0$ might be encoding a current saturation limit on an actuator, say, torque or thrust; here, small deviations are tolerable, but large deviations can become dangerous. As another example, in robot trajectory planning, the inequality might represent a signed distance to a harmful region. Here, violating the constraint means the robot is at risk, a risk that might not scale linearly in the sense that every extra unit of penetration does not cost the same and might be best accounted for by a quadratic increase. Yet another example might occur in a vehicle following a mobile target. Here, the constraint might model the wish to stay within a given radius of the target. If the distance becomes too large, the vehicle might lose sight of the target and fail the tracking mission.

## 2 PRELIMINARIES

We summarize the technical tools used in the analysis of CLASP and state our assumptions.

**Projection operators and distance functions.** For a non-empty, closed convex set $S \subset \mathbb{R}^n$, we let $P_S : \mathbb{R}^n \to \mathbb{R}^n$ be the associated orthogonal projection operator. Thus, $P_S(u)$ is the projection of the point $u$ onto the set $S$ and denotes the point in $S$ that is closest to $u$ with respect to the Euclidean norm $\|\cdot\|$. Such projectors are firmly non-expansive (FNE) operators, which means they satisfy the inequality

$$\|P_S(u) - P_S(v)\|^2 \leq \|u - v\|^2 - \|(u - P_S(u)) - (v - P_S(v))\|^2, \tag{4}$$

for all $u, v \in \mathbb{R}^n$ (see, e.g., Bauschke & Combettes (2017, Proposition 4.8)). Property (4) implies the popular non-expansiveness (NE) property, $\|P_S(u) - P_S(v)\| \leq \|u - v\|$.

We let $d_S : \mathbb{R}^n \to \mathbb{R}$ denoted the associated distance function, $d_S(u) = \min \{\|v - u\| : v \in S\}$. Thus, $d_S(u) = \|u - P_S(u)\|$ and, from (4),

$$\|P_S(u) - v\|^2 \leq \|u - v\|^2 - d_S(u)^2, \tag{5}$$

for all $u \in \mathbb{R}^n$ and $v \in S$.

Finally, the function $d_S$ is Lipschitz continuous with constant 1, that is, $|d_S(u) - d_S(v)| \leq \|u - v\|$ for all $u, v \in \mathbb{R}^n$ (see, e.g., Bauschke & Combettes (2017, page 59, Chapter 4)).

**Convex and strongly convex functions.** Let $h : \mathbb{R}^n \to \mathbb{R} \cup \{+\infty\}$ be an extended real-valued function. It is said to be *proper* if its domain, $\mathrm{dom}\, h = \{x \in \mathbb{R}^n : h(x) < +\infty\}$, is a non-empty set; it is *closed* if each lower level set $\{x \in \mathbb{R}^n : h(x) \leq \alpha\}$ at height $\alpha \in \mathbb{R}$ is a closed set; and it is *convex* if there exists an $m \geq 0$ such that

$$h((1 - \lambda)u + \lambda v) \leq (1 - \lambda)h(u) + \lambda h(v) - \frac{m}{2}\lambda(1 - \lambda)\|u - v\|^2, \tag{6}$$

for all $\lambda \in (0, 1)$ and $u, v \in \mathbb{R}^n$. If $m$ can be taken to be strictly positive, then $h$ is said to be *m-strongly convex*.

If $h$ is a proper, closed, and convex function, then

$$h(v) \geq h(u) + \langle \nabla h(u), v - u \rangle + \frac{m}{2}\|v - u\|^2, \tag{7}$$

for all $u, v \in \mathbb{R}^n$, where, from now on, $\nabla h(u)$ denotes a sub-gradient of $h$ at $u$. ($\langle \cdot, \cdot \rangle$ is the usual inner-product). If, furthermore, $h$ is $m$-strongly convex, then it has a unique global minimizer, say, $u^\star$, (see, e.g., Bauschke & Combettes (2017, Corollary 11.16)); the zero vector is then a sub-gradient of $h$ at $u^\star$, implying via (7) that

$$h(v) \geq h(u^\star) + \frac{m}{2}\|v - u^\star\|^2 \tag{8}$$

for all $v \in \mathbb{R}^n$.

**Assumptions.** Throughout, we impose the following standard COCO assumptions:

**Assumption 1.** *The action set $\mathcal{K}$ is a non-empty, compact convex subset of $\mathbb{R}^n$. It follows that the diameter of $\mathcal{K}$ is upper-bounded by some constant, say, $D \geq 0$, which entails $\|u - v\| \leq D$ for all $u, v \in \mathcal{K}$.*

**Assumption 2.** *The loss functions $f_t : \mathbb{R}^n \to \mathbb{R}$ are convex for all $t \geq 1$, with the magnitude of their sub-gradients bounded by $L$, when evaluated at points in $\mathcal{K}$: $\|\nabla f_t(u)\| \leq L$ for all $u \in \mathcal{K}$ and $t \geq 1$.*

**Assumption 3.** *The constraint functions $g_t : \mathbb{R}^n \to \mathbb{R}$ are convex for all $t \geq 1$, with the magnitude of their sub-gradients bounded by $L$, when evaluated at points in $\mathcal{K}$: $\|\nabla g_t(u)\| \leq L$ for all $u \in \mathcal{K}$ and $t \geq 1$.*

*Moreover, $\mathcal{K} \cap \bigcap_{t=1}^{T} C_t$ is non-empty for all $T \geq 1$ (where $C_t = \{x \in \mathbb{R}^n : g_t(x) \leq 0\}$), so as to ensure the existence of the regret comparator $x_T^\star$—recall (2))*

Assumptions 2 and 3 imply that

$$|f_t(v) - f_t(u)| \leq L \|v - u\| \quad \text{and} \quad |g_t(v) - g_t(u)| \leq L \|v - u\| \tag{9}$$

for all $u, v \in \mathcal{K}$ and $t \geq 1$ (as a consequence, e.g., of the mean-value theorem (Hiriart-Urruty & Lemaréchal, 1993, Theorem 2.3.3)).

**Assumption 4.** *The sequence (of step-sizes) $(\eta_t)_{t \geq 1}$ satisfies $0 < \eta_{t+1} \leq \eta_t$ for all $t \geq 1$. Hence, there exists $\theta > 0$ such that $\eta_t^2 \leq \theta \eta_t$ for all $t \geq 1$ (e.g., take $\theta = \eta_1$).*

## 3 THE CLASP ALGORITHM

In this section, we introduce our algorithm CLASP, which stipulates how the decisions $x_{t+1}$ are made, given the stream of losses $f_t$ and constraints $g_t$ observed up to time $t$. CLASP is a conceptually simple algorithm that, at each iteration (after the first), takes a gradient step with respect to the most recently observed loss function, and then projects onto the most recently observed constraint function, see Algorithm 1.

---

**Algorithm 1** CLASP

**Require:** action set $\mathcal{K}$, horizon $T \geq 1$, step-sizes $\eta_t$ for $1 \leq t \leq T - 1$
    Choose $x_1 \in \mathcal{K}$ and observe $f_1, g_1$
    Accumulate loss $f_1(x_1)$ and penalty $(g_1^+(x_1))^2$
    **for** $t = 1, \ldots, T - 1$ **do**
        Choose $x_{t+1} = \mathcal{P}_{\mathcal{K}_t}(x_t - \eta_t \nabla f_t(x_t))$ and observe $f_{t+1}, g_{t+1}$
        Accumulate loss $f_{t+1}(x_{t+1})$ and penalty $(g_{t+1}^+(x_{t+1}))^2$
    **end for**

---

Here, $\nabla f_t(x_t)$ represents a sub-gradient of $f_t$ at $x_t$. The projection step in CLASP is onto the set $\mathcal{K}_t = \mathcal{K} \cap C_t$, with $C_t = \{x \in \mathbb{R}^n : g_t(x) \leq 0\}$ (as defined in (2)). Note that the set $\mathcal{K}_t$ is non-empty, as per Assumption 3, and that this projection, typically realized as a convex optimization problem, sets CLASP apart from the projection-free methods in Section 1.1. The step-sizes $\eta_t$ are chosen according to whether the loss functions are convex or strongly convex and are specified further ahead in Section 3.2. We write $\widehat{x}_{t+1} = x_t - \eta_t \nabla f_t(x_t)$, and rewrite CLASP as $x_{t+1} = p_{\mathcal{K}_t}(\widehat{x}_{t+1})$ for $t \geq 1$.

We now analyze CLASP, finding upper-bounds for both Regret$_T$ (see (1) and (2)) and the hard constraint violation CCV$_{T,2}$ (see (3)).

Our analysis is modular, one module bounding the metric CCV$_{T,2}$ (Section 3.1) and the other module bounding the regret (Section 3.2).

### 3.1 BOUNDING CCV$_{T,2}$

In this section, we present the module of our analysis that is dedicated to bounding the metric CCV$_{T,2}$. More precisely, we start by finding a generic bound on CCV$_{T,2}$ that is phrased in terms of the length of the step-sizes $\sum_{t=1}^{T} \eta_t$. Our bound, the end result of a progression of three lemmas, is stated

precisely in Lemma 3. The lemmas hold under Assumptions 1 to 4, and their proofs are provided in the Appendix.

**Lemma 1.** *In the COCO setting, for adversarially chosen convex loss functions $f_t$ and convex constraint functions, $g_t$, consider Assumptions 1-4 hold. Then for Algorithm 1, there holds*
$$\sum_{t=1}^{T} d_{\mathcal{K}_t}(\widehat{x}_{t+1})^2 \leq O\left(\sum_{t=1}^{T} \eta_t\right).$$

Lemma 1 bounds the cumulative squared-distance of the intermediate step $\widehat{x}_{t+1}$ (which can be sort of a virtual decision) to the convex set $\mathcal{K}_t$.

The next Lemma 2 leverages this result and obtains a similar bound, but now relative to the decision $x_t$.

**Lemma 2.** *In the COCO setting, for adversarially chosen convex loss functions $f_t$ and convex constraint functions, $g_t$, consider Assumptions 1-4 hold. Then for Algorithm 1, there holds*
$$\sum_{t=1}^{T} d_{\mathcal{K}_t}(x_t)^2 \leq O\left(\sum_{t=1}^{T} \eta_t\right).$$

**Lemma 3.** *In the COCO setting, for adversarially chosen convex loss functions $f_t$ and convex constraint functions, $g_t$, consider Assumptions 1-4 hold. Then for Algorithm 1, there holds $CCV_{T,2} \leq O\left(\sum_{t=1}^{T} \eta_t\right).$*

Lemma 3, the main result of this section, connects the growth of the $CCV_{T,2}$ metric with the growth of the stepsizes. The next section chooses these stepsizes for the convex and strongly convex setting and bounds the regret, completing our analysis.

## 3.2 Analysis of CLASP

Having assembled the necessary elements, we are ready to fully analyze CLASP. The analysis is reported in Theorem 1 for convex losses, and in Theorem 2 for strongly convex ones.

**Convex losses.** We suppose that assumptions 1 to 4 hold. Let $\beta$ be any desired value in $(0,1)$ and, accordingly, set the step-sizes to
$$\eta_t = 1/t^\beta, \quad \text{for } t \geq 1, \tag{10}$$
which complies with Assumption 4. The following Theorem 1 bounds the regret and the CCV of CLASP for this setting.

**Theorem 1** (Convex losses). *In the COCO setting, for adversarially chosen convex loss functions $f_t$ and convex constraint functions, $g_t$, consider Assumptions 1-4 hold. Let $\eta_t = 1/t^\beta$, with $\beta \in (0,1)$. Then, Algorithm 1 achieves the following regret and $CCV_{T,2}$ bounds:*
$$Regret_T \leq O(T^{\max\{\beta, 1-\beta\}}) \quad and \quad CCV_{T,2} \leq O\left(T^{1-\beta}\right).$$

*Proof.* We start by analyzing the $Regret_T$. Express the projection step of CLASP as
$$x_{t+1} = \arg\min_{u \in \mathbb{R}^n} h_t(u),$$
where
$$h_t(u) = \langle \nabla f_t(x_t), u - x_t \rangle + \frac{1}{2\eta_t} \|u - x_t\|^2 + \delta_{\mathcal{K}_t}(u),$$
with $\delta_{\mathcal{K}_t} : \mathbb{R}^n \to \mathbb{R} \cup \{+\infty\}$ denoting the indicator function of the convex set $\mathcal{K}_t$ (i.e., $\delta_{\mathcal{K}_t}(u) = 0$ if $u \in \mathcal{K}_t$, and $\delta_{\mathcal{K}_t}(u) = +\infty$, if $u \notin \mathcal{K}_t$).

Note that $h_t$ is a proper, closed, and $\sigma$-strongly convex function with modulus $\sigma = 1/\eta_t$ and global minimizer $x_{t+1}$. Thus, from (8), we have
$$h_t(x_T^\star) \geq h_t(x_{t+1}) + \frac{1}{2\eta_t} \|x_T^\star - x_{t+1}\|^2. \tag{11}$$

Observing that both $x_{t+1}$ and $x_T^\star$ are points in $\mathcal{K}_t$ (which implies $\delta_{\mathcal{K}_t}(x_{t+1}) = \delta_{\mathcal{K}_t}(x_T^\star) = 0$), we can re-arrange (11) as
$$\frac{1}{2\eta_t}\|x_{t+1} - x_T^\star\|^2 \leq \frac{1}{2\eta_t}\|x_t - x_T^\star\|^2 + \langle \nabla f_t(x_t), x_T^\star - x_t \rangle - \frac{1}{2\eta_t}\|x_t - x_{t+1}\|^2$$
$$+ \langle \nabla f_t(x_t), x_t - x_{t+1} \rangle. \tag{12}$$

We now use the easily-checked fact

$$\max\left\{ -\frac{1}{2\eta}\|u\|^2 + \langle v, u\rangle \,:\, u \in \mathbb{R}^n \right\} = \frac{\eta}{2}\|v\|^2,$$

which holds for all $\eta > 0$ and $v \in \mathbb{R}^n$, and amounts simply to compute the peak value of the concave quadratic function inside the braces. Using this fact with $\eta = \eta_t$ and $v = \nabla f_t(x_t)$, we can bound the sum of the last two terms on the right-hand side of (12), thereby arriving at

$$\frac{1}{2\eta_t}\|x_{t+1} - x_T^\star\|^2 \leq \frac{1}{2\eta_t}\|x_t - x_T^\star\|^2 + \langle \nabla f_t(x_t), x_T^\star - x_t\rangle + \frac{\eta_t}{2}\|\nabla f_t(x_t)\|^2. \quad (13)$$

The convexity of $f_t$ implies $\langle \nabla f_t(x_t), x_T^\star - x_t\rangle \leq f(x_T^\star) - f(x_t)$ (e.g., see (7) with $m = 0$), which, when plugged in (13), yields

$$f_t(x_t) - f_t(x_T^\star) \leq \frac{1}{2\eta_t}\|x_t - x_T^\star\|^2 - \frac{1}{2\eta_t}\|x_{t+1} - x_T^\star\|^2 + \frac{\eta_t}{2}\|\nabla f_t(x_t)\|^2. \quad (14)$$

For $t = 1$, inequality (14) implies

$$f_1(x_1) - f_1(x_T^\star) \leq \frac{1}{2\eta_1}D^2 - \frac{1}{2\eta_1}\|x_2 - x_T^\star\|^2 + \frac{\eta_1}{2}\|\nabla f_1(x_1)\|^2, \quad (15)$$

where Assumption 1 was used to bound the first term in the right-hand side of (14). For $t \geq 2$, we have

$$\frac{1}{2\eta_t}\|x_t - x_T^\star\|^2 = \frac{1}{2\eta_{t-1}}\|x_t - x_T^\star\|^2 + \left(\frac{1}{2\eta_t} - \frac{1}{2\eta_{t-1}}\right)\|x_t - x_T^\star\|^2$$

$$\leq \frac{1}{2\eta_{t-1}}\|x_t - x_T^\star\|^2 + \left(\frac{1}{2\eta_t} - \frac{1}{2\eta_{t-1}}\right)D^2, \quad (16)$$

where the last inequality uses Assumptions 1 and 4.

Plugging (16) in (14) gives

$$f_t(x_t) - f_t(x_T^\star) \leq \frac{1}{2\eta_{t-1}}\|x_t - x_T^\star\|^2 - \frac{1}{2\eta_t}\|x_{t+1} - x_T^\star\|^2 + \frac{\eta_t}{2}\|\nabla f_t(x_t)\|^2$$

$$+ \left(\frac{1}{2\eta_t} - \frac{1}{2\eta_{t-1}}\right)D^2, \quad (17)$$

which is valid for $t \geq 2$. Finally, summing (15) to (17) (instantiated from $t = 2$ to $T$) yields

$$\mathrm{Regret}_T \leq \frac{1}{2\eta_T}D^2 + \sum_{t=1}^{T} \frac{\eta_t}{2}\|\nabla f_t(x_t)\|^2$$

$$\leq \frac{T^\beta D^2}{2} + \frac{L^2}{2}\sum_{t=1}^{T}\frac{1}{t^\beta}$$

$$\leq O(T^{\max\{\beta, 1-\beta\}}),$$

where the second inequality follows from Assumption 2 and (10).

We now turn to the analysis of $\mathrm{CCV}_{T,2}$: combining Lemma 3 with (10) gives directly $\mathrm{CCV}_{T,2} \leq O(T^{1-\beta})$ and concludes the proof. $\qquad\square$

**Strongly convex losses.** We suppose assumptions 1 to 4 are still in force and further assume that there exists $m > 0$ such that each convex loss $f_t$ is $m$-strongly convex for $t \geq 1$. The step-sizes are set to

$$\eta_t = 1/(mt), \quad \text{for } t \geq 1, \quad (18)$$

thereby satisfying Assumption 4.

**Theorem 2** (Strongly convex losses). *In the COCO setting, for adversarially chosen $m$-strongly-convex loss functions $f_t$ and convex constraint functions, $g_t$, consider Assumptions 1-4 hold. Let $\eta_t = 1/(mt)$. Then, Algorithm 1 achieves the following regret and $CCV_{T,2}$ bounds:*

$$Regret_T \leq O(\log T) \quad and \quad CCV_{T,2} \leq O(\log T).$$

*Proof.* Looking first at $\mathrm{Regret}_T$ and replaying the opening moves in the proof of Theorem 1, we may assume that inequality (13) holds, an inequality that we reproduce here for convenience of the reader:

$$\frac{1}{2\eta_t}\|x_{t+1} - x_T^\star\|^2 \le \frac{1}{2\eta_t}\|x_t - x_T^\star\|^2 + \langle \nabla f_t(x_t), x_T^\star - x_t \rangle + \frac{\eta_t}{2}\|\nabla f_t(x_t)\|^2. \qquad (19)$$

The strong convexity of $f_t$ implies, via (7),

$$\langle f_t(x_t), x_T^\star - x_t \rangle \le f_t(x_T^\star) - f_t(x_t) - (m/2)\|x_t - x_T^\star\|^2,$$

which, when inserted in (19), gives

$$f_t(x_t) - f_t(x_T^\star) \le \frac{m(t-1)}{2}\|x_t - x_T^\star\|^2 - \frac{mt}{2}\|x_{t+1} - x_T^\star\|^2 + \frac{\eta_t}{2}\|\nabla f_t(x_t)\|^2 \qquad (20)$$

(where we used (18) to replace $\eta_t$ in some terms). Adding (20) from $t = 1$ to $T$ yields

$$\mathrm{Regret}_T \le \sum_{t=1}^{T} \frac{\eta_t}{2}\|\nabla f_t(x_t)\|^2$$

$$\le \frac{L^2}{2}\sum_{t=1}^{T}\frac{1}{mt}$$

$$= O(\log T),$$

where the second inequality is due to Assumption 2 and (18).

It remains to account for $\mathrm{CCV}_{T,2}$. For this, concatenate Lemma 3 with (18) to get $\mathrm{CCV}_{T,2} \le O(\log T)$, thereby closing the proof.

$\square$

## 4 EXTENSIONS

We now briefly mention two straightforward extensions of CLASP: one to accommodate multiple constraints; the other to handle persistent constraints.

**Multiple dynamic constraints.** In deriving CLASP, Algorithm 1, we assumed that only one constraint function $g_t$ is revealed at iteration $t$. In some applications, however, several constraints might be revealed, say, $M$ constraint functions, denoted $g_{t;m}$ for $1 \le m \le M$. Accordingly, CCV might be adjusted to

$$\mathrm{CCV}_{T,2}^1 = \sum_{t=1}^{T}\sum_{m=1}^{M}\left(g_{t;m}^+(x_t)\right)^2 \quad \text{or} \quad \mathrm{CCV}_{T,2}^\infty = \sum_{t=1}^{T}\max\left\{(g_{t;m}^+(x_t))^2 : 1 \le m \le M\right\}.$$

Adapting CLASP to this setting is easy: it suffices to define the set $\mathcal{K}_t$ as $\mathcal{K} \cap C_{t;m_t}$, where $C_{t;m} = \{x \in \mathbb{R}^n : g_{t;m}(x_t) \le 0\}$ and $m_t$ denotes the index of the constraint incurring the largest violation at $x_t$, and correspondingly upgrade Assumption 3 so as to suppose the existence of $x_T^\star \in \mathcal{K} \cap \mathcal{C}_T$, with $\mathcal{C}_T = \cap_{t=1}^{T}\cap_{m=1}^{M}C_{t;m}$. It can be readily verified that the proofs of Lemmas 1 to 3 remain valid. In particular, Lemma 3 now asserts that $\sum_{t=1}^{T}(g_{t;m}^+(x_t))^2 \le O\left(\sum_{t=1}^{T}\eta_t\right)$ for any fixed $m \in \{1, \ldots, M\}$. This and the fact that there is a finite number $M$ of constraints imply that the statement of Lemma 3 continues to hold when $\mathrm{CCV}_{T,2}$ is replaced by $\mathrm{CCV}_{T,2}^1$ or $\mathrm{CCV}_{T,2}^\infty$. Owing to the modular structure of the CLASP proof, which treats regret and CCV separately, Theorems 1 and 2 remain valid as well. To conclude, the theoretical guarantees of CLASP are preserved.

**Persistent constraints.** The canonical interpretation in the COCO literature about dynamic constraints is that they are transient: the constraint $g_t$ should be satisfied by decision $x_t$, but not necessarily by future decisions $x_{t+1}, x_{t+2}, \ldots$. This interpretation can be read in the common definitions of CCVs in that they penalize only the violation of $x_t$ induced by $g_t$—if that was not the case, if constraints were to be interpreted as persistent, then the definition of a CCV would have to keep the

score of violations of $x_t$ induced by the *history* of constraints $g_1, g_2, \ldots, g_t$ so far, say, through the much more stricter metric

$$\text{CCV}_{T,2}^{\text{hist}} = \sum_{t=1}^{T} \sum_{\tau=1}^{t} (g_\tau^+(x_t))^2. \tag{21}$$

The underlying interpretation of dynamic constraints as being transient can also be inferred from standard COCO algorithms, in the sense that they generate the decision $x_t$ without worrying, let alone enforcing, that it satisfies previously revealed constraints (in this regard, the only exception that we know of is Vaze & Sinha (2025), though the authors do not articulate that such property is built-in in their algorithm to address explicitly persistent constraints).

In any case, although we are not aware of specific applications that are best modeled by persistent constraints, CLASP can be effortlessly adjusted to such settings if needed by defining $\mathcal{K}_t = \mathcal{K} \cap \bigcap_{t=1}^{T} C_t$, with $C_t = \{x \in \mathbb{R}^n : g_t(x) \leq 0\}$. This forces $x_t$ to comply with all constraints revealed so far, that is, $g_\tau(x_t) \leq 0$ holds for $\tau < t$, in which case $\text{CCV}_{T,2}^{\text{hist}}$ collapses back to $\text{CCV}_{T,2}$, and the theoretical analysis goes through unchanged, keeping the guarantees of CLASP intact.

## 5 NUMERICAL EXPERIMENTS

In this section, we evaluate the performance of our proposed algorithm CLASP with the AdaGrad-based algorithm of Sinha & Vaze (2024), the RECOO algorithm of Guo et al. (2022), the Frank-Wolfe-based algorithm of Wang et al. (2025), and the Switch algorithm of Vaze & Sinha (2025). We report results for the cumulative loss and for the violation metrics $\text{CCV}_{T,1}$ and $\text{CCV}_{T,2}$.

**On evaluating $\text{CCV}_{T,2}$.** In line with prior COCO work, we measure $\text{CCV}_{T,2}$ *a posteriori* to illustrate how this theoretically motivated quantity behaves in practice.

Further synthetic and real-data experiments can be found in Appendix A.4. All results were averaged over 100 trials and are reported with 95% confidence intervals [1]. The experiments were performed on a 2020 MacBook Air 13", with an 8-core Apple M1 processor and 16 GB of RAM.

### 5.1 ONLINE LINEAR REGRESSION

This experiment is similar to the one presented by Guo et al. (2022) for the setting of adversarial constraints with a synthetic dataset. At each iteration $t$, the loss function is given by $f_t(x) = \|H_t x - y_t\|^2$, with $H_t \in \mathbb{R}^{4 \times 10}$, $x \in \mathbb{R}^{10}$ and $y_t \in \mathbb{R}^4$, where $H_t(i,j) \sim U(-1,1)$, $1 \leq i \leq 4$, $1 \leq j \leq 10$, and $y_t(i) = H_t(i)\mathbf{1} + \epsilon_i$, such that $\mathbf{1} := (1, \ldots, 1) \in \mathbb{R}^{10}$ and $\epsilon_i$ denotes the standard normal random variable. On the other hand, the constraint function is given by $g_t(x) = A_t x - b_t$, with $A_t \in \mathbb{R}^{4 \times 10}$ and $b_t \in \mathbb{R}^4$, where $A_t(i,j) \sim U(0,2)$ and $b_t(i) \sim U(0,1)$, for $1 \leq i \leq 4$ and $1 \leq j \leq 10$. Note that $g_t$ is a vector-valued map taking values in $\mathbb{R}^4$, and thus corresponds to multiple constraints of the form $g_{t,m}(x) \leq 0$, for $1 \leq m \leq 4$. However, we can replace it with a single scalar function, which is the pointwise maximum of the constraints $g_t(x) = \max\{g_{t,m}(x) : 1 \leq m \leq 4\}$ (see Section 4 for details). The static decision set is given by $\mathcal{K} = \{x \in \mathbb{R}^{10} : 0 \leq x_i \leq 1, \text{for } i = 1, \ldots, 10\}$. The vector $x_1 \in \mathbb{R}^n$ in all algorithms was initialized such that $x_i \sim U(0,1)$, for $1 \leq i \leq n$, where $U(a,b)$ stands for the uniform distribution supported on the interval $(a,b)$.

Figure 1 reports cumulative loss, linear violation ($\text{CCV}_{T,1}$), and squared violation ($\text{CCV}_{T,2}$) over iterations. The AdaGrad-based method (blue) attains the lowest cumulative loss, but only by incurring much larger constraint violations. By contrast, CLASP (orange) controls both $\text{CCV}_{T,1}$ and $\text{CCV}_{T,2}$ at levels comparable to RECOO (green), while remaining simpler and more memory-efficient. The Frank-Wolfe-based algorithm (red) achieves similar cumulative loss as CLASP, RECOO, and Switch, at the cost of higher cumulative constraint violation. Switch (purple) achieves the smallest overall violations, but at the cost of higher per-iteration complexity. Notably, on the squared violation metric $\text{CCV}_{T,2}$—the focus of our analysis—CLASP is competitive with the best-performing baselines, confirming in practice that it can keep violation severity sublinear while maintaining reasonable regret.

---

[1]The source code for the numerical experiments can be found in `https://anonymous.4open.science/r/CLASP-45D2`

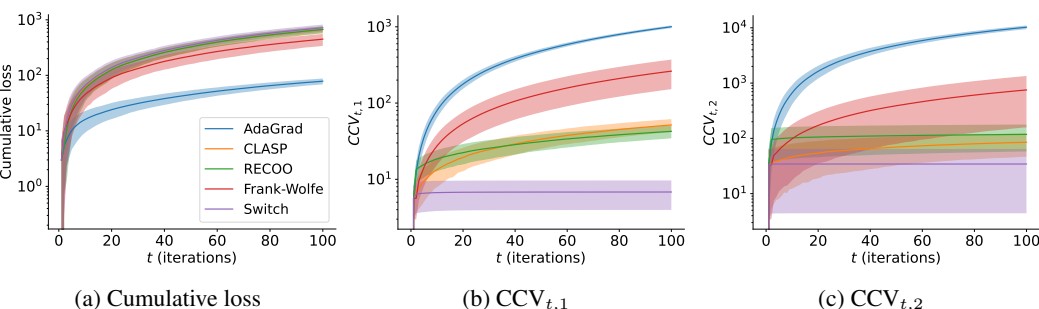

|                     |                     |                     |
| :-----------------: | :-----------------: | :-----------------: |
| (a) Cumulative loss | (b) $\text{CCV}_{t,1}$ | (c) $\text{CCV}_{t,2}$ |

Figure 1: Online linear regression with adversarial constraints. We report (a) cumulative loss, (b) linear violation $\text{CCV}_{T,1}$, and (c) squared violation $\text{CCV}_{T,2}$. AdaGrad achieves the lowest loss but at the cost of very large constraint violations. CLASP controls both violation metrics competitively with RECOO, while remaining more memory-efficient than Switch. Frank-Wolfe attains higher cumulative constraint violation compared with CLASP, RECOO, and Switch while not attaining relevant reduction in the cumulative loss. Switch attains the smallest violations overall but with higher per-iteration complexity. All $\text{CCV}_{T,2}$ values are reported *a posteriori*.

## 6    CONCLUSIONS

**Limitations and Future Work.**    While CLASP attains state-of-the-art guarantees for the squared cumulative constraint violation $CCV_{T,2}$, we do not pursue sharp bounds for the linear metric $CCV_{T,1}$ in the convex regime. Obtaining such bounds would require a dedicated analysis of CLASP, rather than a loose Cauchy-Schwarz conversion, and constitutes a natural continuation of the modular FNE-based framework developed here. A second direction concerns projection-free variants of CLASP, which would broaden its applicability in domains where projections onto $\mathcal{K} \cap C_t$ are computationally costly. We regard both extensions as promising and complementary avenues for future research.

**Conclusions.**    We introduced CLASP, an online COCO algorithm that handles convex losses and dynamic constraints. CLASP aims at minimizing both loss regret and cumulative constraint violation (CCV). In this work, the metric is the squared Cumulative Constraint Violation to account for large violations. For general convex losses, CLASP offers a tunable trade-off between regret and CCV, matching the best performance of previous works designed for static constraints. More importantly, for strongly convex losses, CLASP universally achieves logarithmic bounds on both regret *and* CCV—an advance that, to the extent of our knowledge, is established here for the first time. Algorithmically, CLASP consists of just a gradient step followed by a projection at each iteration, while its analysis is simplified by leveraging the firmly non-expansiveness property of projections. This yields a modular proof structure that disentangles the treatment of regret from that of CCV. This modularity, in turn, makes extensions of CLASP to other settings (e.g., multiple or persistent constraints) straightforward.

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

# A APPENDIX

## A.1 PROOF OF LEMMA 1

Note that $x_T^\star \in \mathcal{K} \cap \mathcal{C}_T \subset \mathcal{K}_t$, implying $x_T^\star = P_{\mathcal{K}_t}(x_T^\star)$ for $1 \le t \le T$. We have

$$
\begin{aligned}
\|x_{t+1} - x_T^\star\|^2 &= \|P_{\mathcal{K}_t}(\widehat{x}_{t+1}) - P_{\mathcal{K}_t}(x_T^\star)\|^2 \\
&\le \|\widehat{x}_{t+1} - x_T^\star\|^2 - d_{\mathcal{K}_t}(\widehat{x}_{t+1})^2 \\
&= \|x_t - \eta_t \nabla f_t(x_t) - x_T^\star\|^2 - d_{\mathcal{K}_t}(\widehat{x}_{t+1})^2 \\
&= \|x_t - x_T^\star\|^2 - 2\eta_t \langle \nabla f_t(x_t), x_t - x_T^\star \rangle + \eta_t^2 \|\nabla f_t(x_t)\|^2 - d_{\mathcal{K}_t}(\widehat{x}_{t+1})^2 \\
&\le \|x_t - x_T^\star\|^2 + 2\eta_t \|\nabla f_t(x_t)\|\|x_t - x_T^\star\| + \eta_t^2 \|\nabla f_t(x_t)\|^2 - d_{\mathcal{K}_t}(\widehat{x}_{t+1})^2 \\
&\le \|x_t - x_T^\star\|^2 + 2LD\eta_t + L^2\eta_t^2 - d_{\mathcal{K}_t}(\widehat{x}_{t+1})^2,
\end{aligned}
$$

where the first inequality is due to the firm non-expansiveness of the operator $P_{\mathcal{K}_t}$ (see (5)); the second inequality is the Cauchy-Schwarz inequality; and the third inequality follows from Assumptions 1 and 2.

Before proceeding, we would like to indicate the pivotal role played by the firmly non-expansiveness property of the operator $P_{\mathcal{K}_t}$. Indeed, if we used only its weaker non-expansiveness property, the

right-hand side of the first inequality would read as $\|\widehat{x}_{t+1} - x_T^\star\|^2$, the key term $d_{\mathcal{K}_t}(\widehat{x}_{t+1})^2$ no longer present. The disappearance of this term would instantly sever the reasoning of the present lemma; as the subsequent lemmas depend upon it, the entire proof would be invalidated.

Re-arranging the last inequality yields

$$d_{\mathcal{K}_t}(\widehat{x}_{t+1})^2 \leq \|x_t - x_T^\star\|^2 - \|x_{t+1} - x_T^\star\|^2 + 2LD\eta_t + L^2\eta_t^2$$
$$\leq \|x_t - x_T^\star\|^2 - \|x_{t+1} - x_T^\star\|^2 + (2LD + \theta L^2)\eta_t,$$

where the last inequality is due to Assumption 4.

Summing up from $t = 1$ to $T$, we obtain

$$\sum_{t=1}^T d_{\mathcal{K}_t}(\widehat{x}_{t+1})^2 \leq \|x_1 - x_T^\star\|^2 + \sum_{t=1}^T (2LD + \theta L^2)\eta_t = O\left(\sum_{t=1}^T \eta_t\right).$$

## A.2 PROOF OF LEMMA 2

We have

$$d_{\mathcal{K}_t}(x_t) = d_{\mathcal{K}_t}(\widehat{x}_{t+1} + (x_t - \widehat{x}_{t+1}))$$
$$\leq d_{\mathcal{K}_t}(\widehat{x}_{t+1}) + \|x_t - \widehat{x}_{t+1}\|$$
$$= d_{\mathcal{K}_t}(\widehat{x}_{t+1}) + \eta_t \|\nabla f_t(x_t)\|$$
$$\leq d_{\mathcal{K}_t}(\widehat{x}_{t+1}) + \eta_t L,$$

where the first inequality follows from the Lipschitz continuity of the distance function $d_{\mathcal{K}_t}$ (see Section 2), and the second inequality from Assumption 2.

It follows that

$$d_{\mathcal{K}_t}(x_t)^2 \leq 2d_{\mathcal{K}_t}(\widehat{x}_{t+1})^2 + 2\eta_t^2 L^2$$
$$\leq 2d_{\mathcal{K}_t}(\widehat{x}_{t+1})^2 + 2\eta_t \theta L^2,$$

where the first inequality comes from the general fact $(a + b)^2 \leq 2a^2 + 2b^2$ for $a, b \in \mathbb{R}$, and the second inequality from Assumption 4.

Summing up from $t = 1$ to $T$ and using Lemma 1, we obtain

$$\sum_{t=1}^T d_{\mathcal{K}_t}(x_t)^2 \leq 2\sum_{t=1}^T d_{\mathcal{K}_t}(\widehat{x}_{t+1})^2 + 2L^2\theta \sum_{t=1}^T \eta_t$$
$$\leq O\left(\sum_{t=1}^T \eta_t\right).$$

## A.3 PROOF OF LEMMA 3

Note that $|g_t^+(u) - g_t^+(v)| \leq |g_t(u) - g_t(v)|$ for all $u, v \in \mathbb{R}^n$. Hence, $|g_t^+(u) - g_t^+(v)| \leq L\|u - v\|$ for all $u, v \in \mathcal{K}$ due to Assumption 3 and its consequence (9).

It follows that

$$g_t^+(x_t) \leq g_t^+(P_{\mathcal{K}_t}(x_t)) + L\|x_t - P_{\mathcal{K}_t}(x_t)\|$$
$$= Ld_{\mathcal{K}_t}(x_t),$$

since $g_t^+(P_{\mathcal{K}_t}(x_t)) = 0$ because $P_{\mathcal{K}_t}(x_t) \in \mathcal{K}_t = \mathcal{K} \cap \{x \in \mathbb{R}^n : g_t(x) \leq 0\}$ and any point $u \in \mathcal{K}_t$ satisfies $g_t(u) \leq 0$.

Equivalently, we have $g_t^+(x_t)^2 \leq L^2 d_{\mathcal{K}_t}(x_t)^2$. Summing up from $t = 1$ to $T$ and using Lemma 2, we obtain

$$\text{CCV}_{T,2} = \sum_{t=1}^{T} (g_t^+(x_t))^2$$

$$\leq L^2 \sum_{t=1}^{T} d_{\mathcal{K}_t}(x_t)^2$$

$$\leq O\left(\sum_{t=1}^{T} \eta_t\right).$$

### A.4  ADDITIONAL NUMERICAL EXPERIMENTS

In this section, we present additional numerical experiments. In particular, a version of the Online Linear Regression experiment with additional rounds, and an experiment for Online Support Vector Machine. The results were obtained by averaging over 100 trials and reported with a $95\%$ confidence interval.

#### A.4.1  ONLINE LINEAR REGRESSION WITH ADDITIONAL ROUNDS

We now present a version of the Online Linear Regression experiment with a larger number of rounds. As stated in Section 1.2, the Switch algorithm is memory-intensive as the set to which the algorithm performs the projections incorporates all previously revealed constraint functions. Therefore, as the iterations unfold, the set becomes increasingly more complex and the associated projection increasingly more expensive. So, we removed Switch from this experiment, as to be able to investigate a larger number of rounds.

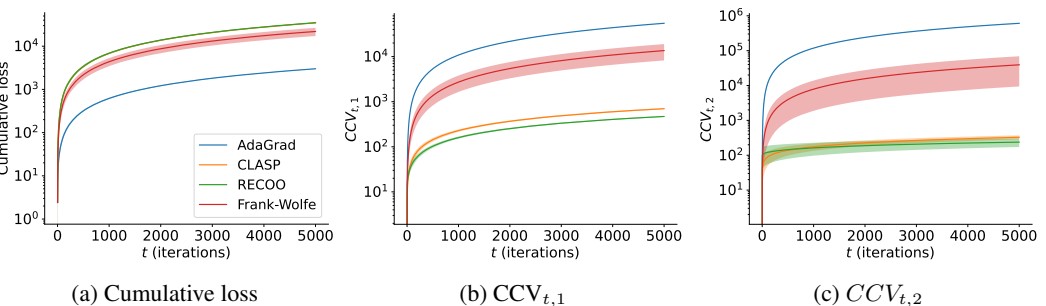

(a) Cumulative loss      (b) $\text{CCV}_{t,1}$      (c) $CCV_{t,2}$

Figure 2: Online Linear Regression with additional rounds.

The results are similar to the ones in Section 5. However, while the Cumulative Constraint Violation (both $\text{CCV}_{T,1}$ and $\text{CCV}_{T,2}$) attained by CLASP and RECOO are very similar, note that, for the $\text{CCV}_{T,2}$ metric, for a larger number of rounds, the CLASP algorithm displays diminished variability.

#### A.4.2  ONLINE SUPPORT VECTOR MACHINE

In our second experiment, we compare the performance of the algorithms for the online update of a Support Vector Machine (SVM) (Boser et al., 1992; Bishop & Nasrabadi, 2006). We consider that, at each iteration $t$, we receive a new labeled sample $(u_t, v_t)$, with $u_t \in \mathbb{R}^P$ the feature vector and $v_t \in \{\pm 1\}$ the label. Thus, at each round $t$, we can formulate the optimization problem in the COCO setting as

$$\underset{x := (w, b) \in \mathcal{K}}{\text{minimize}} \quad \frac{1}{2} \|w\|^2, \quad \text{subject to} \quad -v_t\left(w^T u_t - b\right) + 1 \leq 0,$$

or, in terms of the COCO framework, the revealed loss function is $f_t(x) = \frac{1}{2}\|w\|^2$ and the revealed constraint function is $g_t(x) = -v_t\left(w^T u_t - b\right) + 1$. For this experiment, we use the real-world dataset about wine quality from their physicochemical properties (for the details about the dataset,

see Cortez et al. (2009))[2]. The dataset contains 6497 samples, each sample contains $P = 11$ features, and the quality of the wine is classified between 1 and 9. Some of the features were in $g/dm^3$, while others were in $mg/dm^3$. We ensured that all density features were expressed in $g/dm^3$. In this experiment, we consider the binary classification setting, where we label with $1$ the wine samples with quality equal to or greater than 7, and label with $-1$ the remaining. Thus, we want our classifier to distinguish between high-quality and low-quality wines. In each trial, we reshuffle the dataset so that the order of samples in each trial is always different. The vector $x_1 \in \mathbb{R}^{n+1}$ for all algorithms was initialized such that $x_i \sim U(-1, 1)$, for $1 \leq i \leq n + 1$. The results were obtained by averaging over 100 trials and reported with a $95\%$ confidence interval.

From domain knowledge, we can bound the norm of the feature vector as each physicochemical property has an interval of possible values (for simplicity, we analyzed the possible values encountered in the dataset and concluded that $0 \leq u_t \leq 70$ for all $t$, thus we can use this to define the Lipschitz constant of the constraint functions $L = 70\sqrt{P}$). While the SVM is an unconstrained problem, our algorithms assume that the decision set is compact; thus, based on the constraints on the feature vectors, we define $\mathcal{K} = \{x = (w, b) \in \mathbb{R}^{P+1} : \|x\| \leq 70\sqrt{P}\}$.

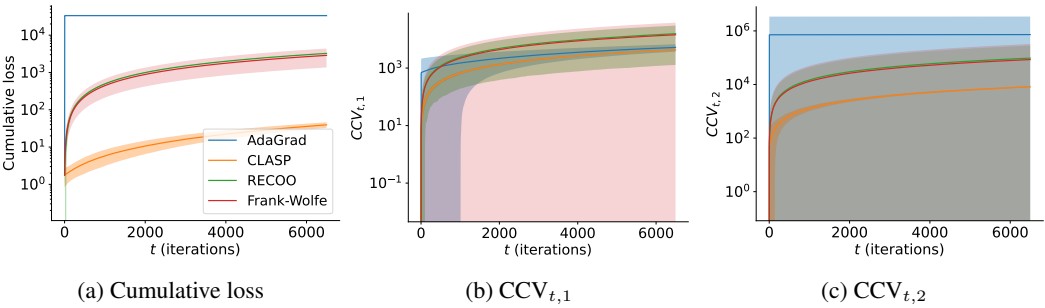

(a) Cumulative loss        (b) $\mathrm{CCV}_{t,1}$        (c) $\mathrm{CCV}_{t,2}$

Figure 3: Online Support Vector Machine

**Remark.** In this experiment, the feasibility assumption is not satisfied, i.e., the set $\mathcal{K} \cap \mathcal{C}_T = \varnothing$, where $\mathcal{C}_T = \bigcap_{t=1}^{T} C_t$. Although this property is used in the analysis of the algorithms in COCO to obtain regret and cumulative constraint violation bounds, most of the algorithms in COCO can still be applied to problems without this property. However, the Switch algorithm, due to the exploitation of nested convex bodies, cannot be applied when the feasibility property is not satisfied. Since, at each iteration, the algorithm projects onto the intersection of past constraint functions, then there will be some $T_0$, with $1 \leq T_0 \leq T$, such that $\mathcal{C}_{T_0} = \bigcap_{t=1}^{T_0} C_t = \varnothing$. Therefore, in this experiment, we cannot compare the performance of the Switch algorithm, but compare our algorithm CLASP with the AdaGrad, RECOO and Frank-Wolfe algorithms.

In Fig. 3, we can visualize both the cumulative loss and the CCV of the different algorithms for each round. In this experiment, the more informative results are in Figs. 3b and 3c, as a constraint violation translates as a transgression of the margin. We see for both constraint violation metrics, $\mathrm{CCV}_{T,1}$ and $\mathrm{CCV}_{T,2}$, the CLASP algorithm achieves better results and with less variability. Furthermore, we see that for the metric $\mathrm{CCV}_{T,2}$, the difference in performance is significantly better.

---

