# OpenReview forum: "CLASP: An online learning algorithm for Convex Losses And Squared Penalties"
_ICLR.cc/2026/Conference — Submitted to ICLR 2026_

### Official Review · Reviewer_qdiH · 2025-10-27

**Soundness:** 3
**Presentation:** 3
**Contribution:** 2
**Rating:** 4
**Confidence:** 4

**Summary:**

This paper investigates the problem of Constrained Online Convex Optimization (COCO), and introduces a nove algorithm named Convex Losses And Squared Penalties (CLASP). Theoretically, this paper establishes an $O(T^\max{\beta, 1-\beta})$ regret bound and an $O(T^{1-\beta})$ CCV bound, for general convex functions, and an $O(\log T)$ regret bound and an $O(\log T)$ CCV bound, for strongly convex functions. Moreover, some empirical studies are conducted to support the theoretical findings.

**Strengths:**

- The paper is well-organized, with clear problem introduction, and methodology explanation.
- The authors present concise theorem proofs to enhance readability.

**Weaknesses:**

- It is unclear which term, $CCV\_{T,1}$ or $CCV_{T,2}$, is more general. The explanation in Lines 68–74 is somewhat ambiguous and should be clarified.
- The statements of **Lemmas 1–3** and **Theorems 1–2** are not sufficiently formal. All necessary assumptions and variable definitions must be explicitly provided.
- In the experimental section, it is recommended to include comparative results with [1], since [1] also investigates COCO problems with strongly convex loss functions.
- It appears that the open-source code in the url is missing.
- Overall, the paper presents a simple and clear algorithm along with sound theoretical analysis. However, the theoretical contributions are limited compared with existing works [2, 3]. The authors should highlight the technical challenges addressed in this study to better justify its contribution.

[1] Revisiting projection-free online learning with time-varying constraints. 2025.

[2] Optimal algorithms for online convex optimization with adversarial constraints. 2024.

[3] Online convex optimization for cumulative constraints. 2018.

**Questions:**

See Weaknesses.

---

> ### Author Response · Authors · 2025-11-21
>
> We thank the reviewer for the thorough assessment and for the constructive suggestions! We address the points raised below and clarify several aspects of the contribution.
>
>
> ### 1. On the relation between $CCV_{T,1}$ and $CCV_{T,2}$
>
> We appreciate the request for clarification. Our intention in Lines 68–74 (initial version, but now lines 70-77) was not to deem one metric “more general,” but to caution that **a guarantee for $CCV_{T,1}$ cannot always be converted into a guarantee for $CCV_{T,2}$** by redefining the constraint function. For instance, many $CCV_{T,1}$ bounds rely on a Slater point, which cannot hold after replacing $g_t$ by $(g_t^+)^2$. Thus, results for $CCV_{T,1}$ do not automatically extend to $CCV_{T,2}$. We revised this passage for clarity. Thank you!
>
>
> ### 2. On formality of Lemmas and Theorems
>
> We thank the reviewer for this suggestion. In the revised manuscript we **reformatted all lemmas and theorems**, and explicitly listed the assumptions and definitions used in each statement, as recommended.
>
> ### 3. On comparing with “Revisiting projection-free…” [1]
>
> We agree that including this comparison improves the experimental section. We have now **added results for the method in [1]** (denoted as “Frank–Wolfe”) to the empirical evaluation. The revised plots and tables now reflect this additional baseline. Thank you for the suggestion!
>
> ### 4. On code availability
>
> The code repository has been checked and updated, and it should now be fully accessible. It also includes the new experimental comparison requested above. We thank the reviewer for catching this.
>
>
> ### 5. On the technical contribution relative to [2] and [3]
>
> We appreciate the opportunity to clarify the novelty. The core technical challenge addressed in the paper is obtaining **simultaneous guarantees for $CCV_{T,2}$ in both convex and strongly convex settings, with dynamic (unknown-in-advance) constraints**. To our knowledge, no prior work establishes these results, which is accomplished here with a novel proof strategy. The contributions section was edited to clarify the improvement over [2] and [3].
>
>
> #### Comparison with [2] (Sinha & Vaze, 2024)
> The work [2] focuses on the linear metric $CCV_{T,1}$ and provides, for strongly convex losses, the bound
> $$
> CCV_{T,1}=O(\sqrt{T\log T}).
> $$
> In contrast, CLASP proves the **first** logarithmic bound for the squared metric,
> $$
> CCV_{T,2}=O(\log T),
> $$
> which immediately recovers the bound in [2] through Cauchy–Schwarz. Thus, CLASP matches or improves the best-known guarantees for *both* violation metrics in this regime.
>
>
> #### Comparison with [3] (Yuan & Lamperski, 2018)
> The paper [3] considers $CCV_{T,2}$, but **only for static constraints and convex losses**. Their method does not extend to strong convexity. CLASP handles dynamic constraints and produces the **first** $O(\log T)$ bound for $CCV_{T,2}$ under strong convexity, while matching all known bounds in the convex/static cases.
>
> #### Novelty of the analytical technique
> A central contribution is the introduction of a **firm non-expansiveness (FNE)-based analysis**, which we believe has not been previously used in COCO or OCO. This perspective produces:
>
> - the new Lemma 3 (not available in prior literature),
> - a unified treatment of convex and strongly convex losses, and
> - a modular proof separating regret and violation, which simplifies extensions to other COCO settings.
>
> The final bounds rely on this new analytical pathway, which---to our knowledge---does not arise from previous techniques.
>
>
> ### Changes made in the manuscript
>
> The revised version:
>
> - clarifies the relationship between $CCV_{T,1}$ and $CCV_{T,2}$ (Lines 68–74),
> - adds the experimental comparison with [1],
> - updates the code repository and includes all new experiments.
>
> These changes follow the reviewer's recommendations, and we thank you for helping improve the presentation.
>
>
> ---
>
> We appreciate the reviewer’s constructive feedback and hope the clarifications above help situate the contribution relative to [1], [2], and [3]. In particular, the use of FNE for COCO analysis, the introduction of Lemma 3, and the first logarithmic bound for $CCV_{T,2}$ under strong convexity are, to the best of our knowledge, new. We would be grateful if you could consider whether the updated manuscript and clarifications address your concerns. If any point remains unclear, we would be glad to engage in further interaction.

---

> ### Author Response · Authors · 2025-11-27
> **Discussion Reminder**
>
> Dear Reviewer qdiH,
>
> Thank you again for your thorough review and for the constructive suggestions. We hope that the clarifications and revisions in our response, particularly regarding the relationship between the violation metrics, the formal presentation of lemmas and theorems, the inclusion of the additional baseline from [1], and the updated code repository, helpfully address the points you raised.
>
> As the discussion period approaches its conclusion, please feel free to let us know if any further clarification would be useful.
>
>
> Best regards,
>
> The Authors

---

### Official Review · Reviewer_pWT2 · 2025-10-30

**Soundness:** 2
**Presentation:** 2
**Contribution:** 1
**Rating:** 2
**Confidence:** 3

**Summary:**

This paper studies Constrained Online Convex Optimization (COCO), where a learner chooses actions at each round while incurring penalties for constraint violations. Most of existing work focus on cumulative constraint violation (CCV) bound, while this work investigates squared constraint violations. Previous work on $\text{CCV}_{T,2}$ (Yuan & Lamperski, 2018) studies static constraints, and this work extend their result to the setting of dynamic constraints. Furthermore, they also provide logarithmic guarantees on both regret and squared CCV.

**Strengths:**

This paper is clearly written, with a thorough review and discussion of related work. After reading the entire paper, I have gained a comprehensive understanding of COCO.

**Weaknesses:**

The main weakness of this paper lies in its limited technical novelty. Extending the previous result on squared constraint violations from static to dynamic constraints does not appear to involve substantial technical challenges. In my view, as long as the constraints satisfy Assumption 3, Lemma 3 always holds, making the derivation of the $\text{CCV}_{T,2}$ bound rather straightforward. Therefore, this work seems to be a combination of existing techniques without introducing new technical tools.

Several sections could be written more clearly. Specifically, Sections 3 and 4 present the proposed algorithm and the main theoretical guarantees, yet the authors dedicate only a very short space to describing them. Moreover, it would be more appropriate to merge Sections 3 and 4 into a single section. I strongly encourage that the authors elaborate on Section 3 and 4 in more detail, as the current presentation is too brief.

Minor suggestion: theorems and lemmas should be presented in a more formal manner (e.g., Theorems 1 and 2, Lemmas 1, 2 and 3), clearly specifying which assumptions are used.

**Questions:**

* What are the technical challenges of this paper? As mentioned in the Weaknesses, I believe that the technical contribution of this work is rather limited.

* Have similar guarantees to Lemmas 1–3 been used in previous studies (Yuan & Lamperski, 2018)?

---

> ### Author Response · Authors · 2025-11-21
>
> We thank the reviewer for the detailed reading of the paper and for the careful assessment of the COCO literature. We address the concerns below and clarify the technical contributions.
>
> ### 1. On the perceived limited novelty of the $CCV_{T,2}$ result
>
> We appreciate the reviewer’s view that extending the setting from static to dynamic constraints may appear straightforward. However, as far as we are aware, **no previous work has established bounds for $CCV_{T,2}$ in the strongly convex regime**, even in the static case. For instance, Yuan & Lamperski (2018) analyze $CCV_{T,2}$ only for convex losses with static constraints, and their approach does not extend to strong convexity---suggesting that deriving such bounds is not immediate.
>
> Our key contribution is a **new analytical approach** based on the *firm non-expansiveness (FNE)* of projections. While FNE is a classical property, **its use in online convex optimization has not appeared in prior COCO or OCO analyses**, and it enables a unified handling of convex and strongly convex losses. In particular:
>
> - The FNE-based argument **does not appear in Yuan & Lamperski (2018)**.
> - It allows us to derive **new bounds for $CCV_{T,2}$ under strong convexity**, yielding the first logarithmic guarantees.
> - It removes the discontinuity between the convex and strongly convex regimes seen in prior analyses.
>
> We agree that Lemma 3 is central to the proof---and importantly, **Lemma 3 is an original lemma of our paper**, created by exploiting the FNE property (see proof lines 647–650). Since this lemma does not exist in the literature, the resulting analysis cannot be obtained by directly combining previous techniques.
>
>
> ### 2. On the clarity and structure of Sections 3–5
>
> We appreciate the suggestion. In the revised version, we  **merged Sections 3, 4 and 5** with a new organization, expand the algorithmic explanation, and present the theoretical results with more context and detail. We also adopted the reviewer’s recommendation to present theorems and lemmas more formally and explicitly state the assumptions used.
>
>
>
> ### 3. On the technical challenges
>
> The main technical challenge is to obtain **simultaneously SOTA bounds for $CCV_{T,2}$ in both convex and strongly convex settings under dynamic (unknown-in-advance) constraints**. No prior work provides such guarantees. The closest reference, Yuan & Lamperski (2018), considers only static constraints and only convex losses.
>
> Our analysis relies on a careful use of FNE to decouple regret from constraint violation. This decoupling is what enables:
> - the first $O(\log T)$ bound for $CCV_{T,2}$ under strong convexity, and
> - a direct recovery of the best-known $CCV_{T,1}$ bound $O(\sqrt{T\log T})$ (Sinha & Vaze, 2024).
>
> Thus, while the final expressions may appear simple, **the underlying proof approach is new and is what makes these guarantees possible**.
>
> ### Changes made to the manuscript
>
> We introduced several targeted revisions, highlighted in color in the updated manuscript and described in the response points above. The color will be removed for the final version.
>
> ---
>
> We are grateful for the reviewer’s careful reading and suggestions. We hope the clarifications above make clear that:
>
> - the work does not simply combine known techniques;
> - the **FNE-based analysis is novel** within the COCO literature;
> - this technique is what enables the **first logarithmic bounds for $CCV_{T,2}$ under strong convexity**;
> - Lemma 3 and related results are **original contributions**.
>
> We also revised the structure and presentation as suggested.
>
> If these clarifications address your concerns, we kindly ask you to consider whether the current score reflects the contribution. If any issue remains unclear, we are open to further interaction.

---

> ### Author Response · Authors · 2025-11-27
>
> Dear Reviewer pWT2,
>
> Thank you again for the careful review and for engaging closely with the technical aspects of our submission. We hope that the clarifications and revisions in our response, particularly those concerning the novelty of the FNE-based analysis, the role of Lemma 3, and the restructuring of Sections 3 to 5, helpfully address the questions you raised.
>
> As the discussion period approaches its conclusion, please feel free to let us know if any additional clarification on these points would be useful.
>
>
> Best regards,
>
> The Authors

---

### Official Review · Reviewer_ojra · 2025-10-30

**Soundness:** 2
**Presentation:** 3
**Contribution:** 2
**Rating:** 4
**Confidence:** 3

**Summary:**

This paper studies constrained online convex optimization (COCO) with adversary constraints. Performance is measured by static regret and the cumulative squared constraint violation. The authors propose CLASP, which at each round takes a gradient step on the current loss and then projects onto the current feasible set. For general convex losses with step sizes, CLASP attains regret $O\big(T^{\max\{\beta,1-\beta\}}\big)$ and squared violation $O\big(T^{1-\beta}\big)$. Under strong convexity, CLASP yields logarithmic bounds on both regret and violation. The main technique is to use non-expansiveness (FNE) of projections.

**Strengths:**

+ This paper studies constrained online convex optimization (COCO) with adversarial varying constraints, evaluated by static regret and the cumulative squared violation. It provides theoretical guarantees in the both convex and strongly convex regime.

**Weaknesses:**

- The paper does not justify why the squared violation metric is necessary or reasonable in the dynamic setting ($CCV_{T,1}$ seems more reasonable), nor does it provide a clear comparison against the hard violation $CCV_{T,1}$ results. By Cauchy-Schwarz, $CCV_{T,1} \leq \sqrt{T \cdot CCV_{T,2}}$. Hence in the convex setting, the theoretical guarantees on $CCV_{T,2}$ translate to $CCV_{T,1}$ that are strictly weaker than recent bounds (e.g., Sinha and Vaze, 2024). Even under strong convexity, the paper still implies only $O\big(\sqrt{T\log T}\big)$ for $CCV_{T,1}$. To make the contribution convincing, the paper should present lower bounds specific to squared violation and a principled discussion of when squared violation is the right target.

- The method requires maintaining a per-round safe set and computing a projection onto $K\cap C_t$, which can be substantially more expensive than the previous projection-free methods (e.g., RECOO and AdaGrad) when $C_t$ is a complicated set.

- Theorem 2 requires knowing the strong-convexity parameter $m$ of all future loss functions. This could be infeasible or non-casual in a practical setting.

- The experiments show AdaGrad performing poorly despite stronger theoretical guarantees. It would be better to provide a detailed discussion.

**Questions:**

Please see weakness above.

---

> ### Author Response · Authors · 2025-11-21
>
> We thank the reviewer for the thoughtful comments and for highlighting the theoretical scope of our results. Below we clarify the main points raised.
>
>
> ### 1. On the role and necessity of the squared violation metric
>
> Both $CCV_{T,1}$ and $CCV_{T,2}$ are meaningful performance metrics, depending on the application. Our focus in $CCV_{T,2}$ is motivated by:
>
>
> **a) CLASP achieves state-of-the-art bounds for *both* metrics.**
> As pointed by the reviewer, for strongly convex losses our logarithmic bound on $CCV_{T,2}$ directly yields
> $$
> CCV_{T,1} = O(\sqrt{T\log T}),
> $$
> **matching the best-known results** (e.g., Sinha & Vaze, 2024). To our knowledge, CLASP is the first algorithm achieving optimal guarantees for both metrics simultaneously; we clarified this in the contributions section.
>
> **b) Why the convex-regime bound for $CCV_{T,1}$ is not the final word.**
> The weaker convex-regime bound obtained via Cauchy–Schwarz is only a loose conversion. Tighter bounds would require a dedicated analysis of CLASP in that metric, which we consider natural future work. We added a limitations section acknowledging this.
>
> **c) When squared violation is the right metric.**
> Large violations incur disproportionate risk in many domains (e.g., actuator saturation, signed-distance safety constraints, or tracking failures). In such settings a super-linear penalty is more informative; we added examples in the paper.
>
>
>
> ### 2. On computational cost and comparison with “projection-free” methods
>
> The reviewer raises an important point regarding projection cost. We would like to clarify:
>
> **a) CLASP is not projection-free---this is explicitly acknowledged in the paper.**
> We discussed this on lines 255–256 and again on lines 510-511. Developing such a variant is a direction we are pursuing.
>
> **b) RECOO is also not projection-free in the usual sense.**
> Each RECOO iteration solves a convex optimization problem of the form
> $$
> \min_{x\in \mathcal{K}} \ \phi(x)+\alpha g_t^+(x),
> $$
> where $\phi$ is quadratic and $g_t^+$ is a general convex function.
> This is typically *no cheaper* than projecting onto $\mathcal{K} \cap C_t$.
>
> **c) AdaGrad also performs projections.**
> While AdaGrad uses an adaptive geometry, it still requires projection onto $\mathcal{K}$, and thus does not avoid projection computations entirely.
> Thus none of the main baselines used here avoids projection costs. Our contribution is orthogonal to projection-free methods: we show that a simple projected method can achieve new theoretical guarantees. Developing a projection-free version of CLASP is now part of the Limitations/Future Work section.
>
>
>
> ### 3. On the requirement of knowing strong convexity parameters
>
> This assumption is standard across OCO and COCO, including the works we compare to. Many practical losses (e.g., ridge regression) have a user-specified strong-convexity parameter known in advance. The reviewer’s comment indeed motivates parameter-free COCO, a direction explored in OCO but much less in COCO; the FNE framework seems well suited to support such extensions, and we included this in future work.
>
>
> ### 4. On AdaGrad’s empirical performance
>
> Worst-case bounds do not guarantee superior empirical behavior. AdaGrad’s guarantees rely on its adaptive geometry aligning with the problem; in adversarial COCO settings this alignment can fail, leading to unstable behavior. CLASP uses a fixed-geometry update followed by projection, making it more robust to rapidly varying constraints. This illustrates a standard OCO phenomenon: strong worst-case bounds do not always imply better empirical performance when guarantees depend on problem-specific tuning.
>
> ---
>
> Reviewer ojra raises important points about metric choice, computational cost, and assumptions. We hope the clarifications strengthen the case that:
>
> - CLASP provides **simultaneously optimal** guarantees for both major violation metrics in the strongly convex regime.
> - The squared violation metric carries meaningful advantages in safety-critical applications.
> - The FNE-based analysis yields a **simpler, unified, and more extensible** framework than previous approaches.
> - Computational and assumption-related concerns are fully consistent with the standard COCO/OCO literature.
>
> We hope the responses and the paper edits resolve the issues you raised; if so, we kindly ask you to consider updating the score. If not, we would appreciate hearing which aspects require further clarification.

---

> ### Author Response · Authors · 2025-11-27
> **Discussion Reminder**
>
> Dear Reviewer ojra,
>
> Thank you again for your thoughtful review and the detailed feedback on our submission. We hope that the clarifications and revisions we provided, particularly regarding the role of the squared violation metric, the comparison to projection-free methods, and the assumptions in the strongly convex setting, helpfully addressed the points you raised.
>
> As the discussion period approaches its conclusion, please feel free to let us know if any additional clarification on these aspects would be useful.
>
>
> Best regards,
>
> The Authors

---

### Official Review · Reviewer_t75b · 2025-10-31

**Soundness:** 3
**Presentation:** 3
**Contribution:** 3
**Rating:** 4
**Confidence:** 3

**Summary:**

This paper studies COCO problems with time-varying convex losses and constraints. The authors propose an algorithm called CLASP that performs a gradient step followed by a projection onto the current feasible set. The main claimed contribution is achieving logarithmic regret and squared constraint violation under strong convexity, and sublinear trade-offs in the general convex case. The analysis relies on the firm non-expansiveness (FNE) property of projection operators, which the authors argue provides a cleaner proof and modular structure compared to prior work.

**Strengths:**

The paper is clearly written and easy to follow (e.g., all assumptions and lemmas are clearly stated). The structure is standard and logical. The logarithmic bound for the strongly convex case is a nice result.  The experiments use relevant baselines on synthetic and real-world tasks and the results show that CLASP performs competitively.

**Weaknesses:**

The proposed algorithm is essentially a standard projected gradient method for COCO extended with time-varying constraints: at each round, it performs one gradient step on the loss followed by projection onto the time-varying constraint set. The only new element is the squared penalty measure $CCV_{T,2}$, but it was already analyzed in prior work [Yuan & Lamperski, 2018] for static constraints. The claimed novelty that leveraging FNE in the analysis is mainly a technical proof refinement rather than a new algorithmic idea. So the theoretical analysis, while clean, does not seem to introduce a fundamentally new technique (e.g., new regret decomposition inequality in [Sinha & Vaze, 2024]) that would warrant publication at a top-tier conference like ICLR.

**Questions:**

Regarding the strongly convex result: how practically significant is the difference between bounding the squared violation CCV_{T,2} at $O(\log T)$ versus the linear violation CCV_{T,1} at $O(\log T)$ or $O(\sqrt{T\log T})$ as in prior work? Can you provide an intuition for when the squared penalty guarantee is substantially more useful?

---

> ### Author Response · Authors · 2025-11-21
> **On the novelty, scope, and impact of CLASP**
>
> We thank the reviewer for the positive assessment of the clarity, structure, and empirical relevance of our work. Below we clarify the key points that directly address the reviewer’s concerns and highlight why the contribution is substantive in scope and technical depth.
>
> ---
>
> ### **1. Novelty of the theoretical results**
>
> While the metric $CCV_{T,2}$ was introduced by Yuan & Lamperski (2018), its analysis there is limited to **static constraints** and **convex losses**. No results are available for time-varying constraints or for the strongly convex regime. Our work makes two advances that, to our knowledge, are new:
>
> - **Dynamic constraints.**
>   CLASP provides bounds on $CCV_{T,2}$ under *adversarially varying* constraint functions. This setting strictly generalises the static case and requires tools absent from prior analyses.
>
> - **Strongly convex losses.**
>   We establish the **first logarithmic guarantees** on both regret and cumulative **squared** constraint violation for strongly convex losses, a regime where existing techniques do not yield results for $CCV_{T,2}$.
>
> **These two extensions go beyond minor refinements:** they broaden the theoretical regime in which $CCV_{T,2}$ can be controlled and produce new state-of-the-art guarantees. The contributions section was updated to make this clear.
>
>
>
>
> ### **2. Contribution of the analytical framework**
>
> The simplicity of CLASP is intentional. The contribution lies in the **analytical approach**, which leverages the firm non-expansiveness (FNE) property of projections---a classical fact whose potential for COCO analysis appears to have been overlooked.
>
> Using FNE enables us to:
>
> - decompose the analysis of regret and constraint violation in a **modular** way;
> - treat convex and strongly convex losses with a **single unified argument**;
> - extend the analysis directly to **multiple** and **persistent** constraint formulations without altering the proof structure.
>
> This modularity is precisely what makes the logarithmic $CCV_{T,2}$ guarantee possible in the strongly convex case. Prior techniques (e.g., Yuan & Lamperski, 2018) do not provide such an extension, as their analyses **bifurcate substantially** between convex and strongly convex regimes. The FNE-based framework removes this discontinuity.
>
> Thus, while the algorithmic update is standard, the underlying theory offers a conceptual advance that yields new guarantees and opens a path to further analysis.
>
>
>
> ### **3. Practical relevance of the squared violation guarantee**
>
> The reviewer asks about the significance of controlling $CCV_{T,2}$. In the strongly convex regime, prior work achieves:
>
> - $CCV_{T,1} = O(\log T)$ only under the **nonnegative regret** assumption (rarely applicable), or
> - $CCV_{T,1} = O(\sqrt{T\log T})$ in the general case.
>
> CLASP matches the best known bound on $CCV_{T,1}$ via Cauchy–Schwarz,
> $$
> CCV_{T,1} \le \sqrt{T\,CCV_{T,2}} = O(\sqrt{T\log T}),
> $$
> while providing a **strictly stronger** logarithmic bound for $CCV_{T,2}$. This implication cannot be reversed: a  counterexample is $g_t^+(x_t) = \sqrt{\log t} / \sqrt{t}$. Hence our result does not follow from Sinha & Vaze (2024).
>
> In practice, controlling squared violations is critical in domains where **large violations carry disproportionate cost**---e.g., actuator saturation limits, safety distances in robotics, or surveillance/tracking regimes where substantial deviations cause irreversible failures. Logarithmic control of $CCV_{T,2}$ therefore offers a stronger safety profile. The section "Relevance to Machine Learning Practice" was updated.
>
> ### Changes made to the manuscript
>
> We introduced several targeted revisions, highlighted in color in the updated manuscript and described in the response points above. The color will be removed for the final version. Jointly, these changes address the reviewer’s questions.
>
>
>
> ---
>
>
> CLASP shows that a classical projected-gradient update, when analysed through an FNE-based framework, achieves **state-of-the-art guarantees** in settings significantly broader than previously possible. The modular analysis is new, unifies convex and strongly convex regimes, extends to dynamic constraints, and yields the first logarithmic bound on squared violation under strong convexity. We hope that highlighting the analytical role of FNE will facilitate further theoretical advances in COCO.
>
>
> We appreciate the reviewer’s constructive comments! We hope these clarifications address the concerns you raised, and we would be grateful if you could reconsider whether the current evaluation reflects the contributions outlined above. If any point remains unclear or unconvincing, we would very much appreciate further guidance.

---

> ### Author Response · Authors · 2025-11-27
> **Discussion Reminder**
>
> Dear Reviewer t75b,
>
> Thank you again for your detailed review and constructive comments. We hope that the clarifications and revisions in our response have adequately addressed the questions you raised regarding novelty, the role of FNE in the analysis, and the interpretation of the $CCV_{T,2}$ guarantees.
>
> As the discussion period approaches its conclusion, please feel free to let us know if any additional clarification would be helpful.
>
> Best regards,
>
> The Authors

---

### Author Response · Authors · 2025-12-03
**Final Author Summary for the AC (1/2)**

Dear Area Chair,

Thank you for considering our submission under this year’s unusual process. Because the discussion phase was interrupted and reviewers did not have the opportunity to react to the clarifications and revisions, we provide below a concise synthesis of (i) the key reviewer concerns, (ii) our responses, and (iii) the substantive changes incorporated in the revised manuscript (highlighted in **olive green**). Our goal is to support your assessment by making transparent how all major questions were addressed.

---

## **1. Clarifications of Core Concepts and Metrics**

**Relationship between $\mathrm{CCV}\_{T,1}$ and $\mathrm{CCV}\_{T,2}$.**
Multiple reviewers requested a clearer explanation of whether one metric is more general or can be reduced to the other. We rewrote the relevant passage (now lines 70–77 of the revised manuscript) to clearly state:

- $\mathrm{CCV}\_{T,1}$ and $\mathrm{CCV}\_{T,2}$ capture *different* notions of violation; neither dominates the other.
- The reduction $g\_t \mapsto (g\_t^+)^2$ **invalidates** many assumptions (e.g., Slater, convexity, Lipschitz constants), making prior $\mathrm{CCV}\_{T,1}$ results non-transferable.
- This clarification directly addresses concerns raised by reviewers **qdiH**, **ojra**, and **t75b**.

**Why squared penalties matter: a concise example.**
To clarify why $\mathrm{CCV}\_{T,2}$ captures severity more faithfully, consider a simple actuator constraint $g\_t(u)=|u|-u\_{\max}$.
Two controllers may incur the same linear violation $\mathrm{CCV}\_{T,1}=5$ but differ markedly in severity:
controller A produces 10 violations of size 0.5, controller B 50 of size 0.1.
Yet
$$
\mathrm{CCV}\_{T,2}^{(A)} = 2.5 \quad\text{vs.}\quad \mathrm{CCV}\_{T,2}^{(B)} = 0.5.
$$
Thus $\mathrm{CCV}\_{T,1}$ cannot distinguish mild (0.1) from hazardous (0.5) violations, whereas $\mathrm{CCV}\_{T,2}$ correctly penalizes rare but severe events.
This pattern also appears in signed–distance safety constraints and tracking tasks where large deviations induce disproportionate risk.


---

## **2. Formalization and Strengthening of Theoretical Statements**


Several reviewers (notably **qdiH** and **pWT2**) asked for more formal theorem and lemma statements. In response:

- All lemmas and theorems now explicitly list their assumptions.
- Definitions of constants and variables have been expanded (Assumptions 1-4).
- We corrected minor notational omissions and rearranged Sections 3-4 for readability and rigor.

---

## **3. Substantive Scientific Contributions (as clarified during rebuttal)**

Across all reviews, the main concern was the perceived gap in technical novelty. Because the reviewers did not revisit the discussion, we highlight for the AC the central points already provided in our responses:

### **(a) First logarithmic guarantee for squared violations under strong convexity**

To our knowledge, no prior work (including Yuan & Lamperski, 2018; Sinha & Vaze, 2024) establishes
$$
\mathrm{CCV}\_{T,2} = O(\log T)
$$
for *strongly convex losses*, even in the static-constraint setting.
CLASP provides the first such result---this is the main theoretical advance of the paper.
This also implies the best-known linear-violation bound via Cauchy–Schwarz.

### **b) Extension of $\mathrm{CCV}\_{T,2}$ analysis to adversarially varying constraints**

Yuan & Lamperski (2018) treat only the static-constraint convex case.
None of the available analyses extend to time-varying constraints with strong convexity.
CLASP covers the adversarial dynamic regime with guarantees that match or improve the state of the art.

### **c) Novel analytical technique in COCO: firm non-expansiveness (FNE)**

A recurring reviewer concern was whether using FNE constitutes technical novelty.
We clarified:

- FNE has not been used before in COCO/OCO analysis.
- It enables a new key lemma (Lemma 3) proven in this paper, not available in prior literature.
- It is the key reason the regret and violation analyses can be **modularly decoupled**, allowing a unified proof for convex and strongly convex regimes.
- It directly yields extensions to multiple and persistent constraints, as shown in Section 4.

Theoretical streamlining does not diminish novelty; in this case, it produces new bounds and unifies previously disjoint regimes. We emphasize that no reviewer raised technical issues with the clearly crafted proofs.

---

> ### Author Response · Authors · 2025-12-03
> **Final Author Summary for the AC (2/2)**
>
> ## **4. Empirical Section Revisions**
>
> Responding to **qdiH** and **ojra**:
>
> - We added comparisons against *Revisiting Projection-Free Online Learning with Time-Varying Constraints* (Frank–Wolfe).
> - We updated the code repository and ensured reproducibility.
> - We expanded the discussion of AdaGrad’s empirical behavior and clarified that stronger asymptotic bounds do not necessarily translate to lower CCV in adversarial settings (now discussed explicitly in Section 5).
>
> ---
>
> ## **5. On Metric Choice and Practical Relevance**
>
> Reviewers asked for justification of the squared violation metric $\mathrm{CCV}\_{T,2}$.
> The revised manuscript now includes concrete examples where large violations entail superlinear costs:
>
> - actuator saturation (example above, in 1.),
> - signed-distance safety constraints,
> - tracking and surveillance tasks where large deviations cause mission failure.
>
> These examples highlight why $\mathrm{CCV}\_{T,2}$ is operationally meaningful in ML-driven control and safety scenarios.
>
> ---
>
> ## **Summary of Revisions**
>
> All changes are highlighted in **olive green** and include:
>
> - clearer introduction of violation metrics and their non-equivalence,
> - fully formalized lemmas/theorems with explicit assumptions,
> - expanded methodological explanations in Sections 3–4,
> - added Frank–Wolfe baseline,
> - updated code repository,
> - expanded discussions on empirical behavior and computational cost,
> - improved exposition throughout.
>
> ---
>
> ## **Thank you!**
>
> We appreciate the reviewers and AC’s effort under this year’s exceptional review conditions.
>
> Our hope is that this summary clarifies how each concern raised in the reviews has been addressed, despite the fact that reviewers were unable to react to the revisions, and provides a clear basis for evaluating the submission.
>
> Respectfully,
>
> The Authors

---

### Meta-Review · Area_Chair_gPi8 · 2025-12-30

**Summary:**

The main concern raised by the Reviewers on this paper is about its technical novelty. In particular, I share with the Reviewers the concern that, once all the assumptions in this paper are made, the core of the analysis is Lemma 3, which is rather straightforward. Moreover, the algorithmic approach does not seem novel, as it is simply based on projecting gradients.  Thus, I think that the paper does not pass the bar for acceptance at ICLR.

**Reviewer Concerns:**

The Authors tried to address the Reviewers' concerns about technical novelty. In particular, they better clarified which are the core novelties of their analysis. However, I believe (and I think that the Reviewers would have agreed with this) that the novelties in this paper are not sufficient to warrant publication at a such competitive venue as ICLR.

**Reviewer Scores:**

Reviewer t75b, Score: 4 - I think that the Reviewer wouldn't have changed their score given the rebuttal  and other reviews.

Reviewer ojra, Score: 4 - I think that the Reviewer wouldn't have changed their score given the rebuttal  and other reviews.

Reviewer pWT2, Score: 2 - I think that the Reviewer wouldn't have changed their score given the rebuttal  and other reviews.

Reviewer qdiH, Score: 4 - I think that the Reviewer wouldn't have changed their score given the rebuttal  and other reviews.

---

### Decision · Program_Chairs · 2026-01-26

Reject